# Plant Hormone Signals Mediate Melatonin Synthesis to Enhance Osmotic Stress Tolerance in Watermelon Cells

**Manwen Yan** [1,2], **Mingyan Li** [2], **Zhuoying Ding** [1,2], **Fei Qiao** [3] and **Xuefei Jiang** [1,2,*]

1   Sanya Nanfan Research Institute, Hainan University, Sanya 572000, China; cherry9998@163.com (M.Y.)
2   Key Laboratory for Quality Regulation of Tropical Horticultural Crops of Hainan Province/Key Laboratory of Tropical Agritourism in Greenhouse of Haikou, School of Tropical Agriculture and Forestry, Hainan University, Haikou 570228, China; limy233233@163.com
3   Tropical Crops Genetic Resources Institute, Chinese Academy of Tropical Agricultural Sciences, Haikou 571101, China
*   Correspondence: xuefei.jiang@hainanu.edu.cn

**Abstract:** Melatonin, a multifunctional signaling molecule, has been shown to play a significant role in response to abiotic stress. Several species have been reported to unveil melatonin's effect on osmotic stress; however, the signal transduction mechanism of phytohormone-mediated melatonin biosynthesis in plant species remains unclear. In this study, although plants can biosynthesize melatonin, the exogenous application of melatonin to watermelon cells can improve cell growth in response to osmotic stress by regulating the antioxidant machinery of cells. Regarding the melatonin synthesis pathway, ClOMT (ClASMT and ClCOMT) is a multi-gene family, and ClSNAT has two members. Both *ClOMTs* and *ClSNATs* harbor the *cis*-elements in their promoter regions responding to various hormones. Among abscisic acid (ABA), methyl jasmonate (MeJA), and salicylic acid (SA), ABA treatment observably upregulated the expression of *ClOMTs* and *ClSNATs*, and the accumulation of melatonin with ABA treatment reached a level comparable to that with osmotic stress by mannitol treatment. Furthermore, when hormone biosynthesis inhibitors were added to cells before osmotic stress, the expression of *ClOMTs* and *ClSNATs*, as well as melatonin accumulation, were significantly suppressed with the ABA biosynthesis inhibitor. This study demonstrated the crucial role of melatonin biosynthesis in response to osmotic stress via plant hormone signal transduction. It showed that ABA signaling plays a dominant role in melatonin synthesis under osmotic stress.

**Keywords:** *Citrullus lanatus*; osmotic stress; melatonin; plant hormone signals; gene expression

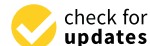



## 1. Introduction

Plants are sessile organisms directly exposed to harsh environmental conditions, including biotic and abiotic stresses [1]. Salt, heat/cold, and drought stress are the most important abiotic stresses; individual or combined stresses can induce osmotic stress, negatively affecting plant reproduction and survival and ultimately limiting plant productivity and yield in agriculture. Since osmotic stress has been associated with the whole evolution process of plants, plants have developed diverse mechanisms to deal with detrimental effects, containing the selectivity of ions, synthesis of organic solutes, alternation in membrane components, and changes in plant hormone levels [2,3]. In this sense, plant hormones, nitric oxide (NO) and hydrogen sulfide, are directly involved in the response to osmotic stress. In recent years, a large number of studies have declared that a pleiotropic molecule, melatonin, plays an essential role in plants responding to abiotic stresses. Exogenously applied melatonin has been proven to have a positive role in reducing the influence of coercion on plants [4].

Melatonin, an essential amino acid derivative, was first isolated from the bovine pineal gland in 1958 [5]. It was not found in higher plants until 1995 [6]. Melatonin has high lipophilic and partially hydrophilic molecular properties, which can easily be

distributed across the cell membrane to the cytoplasm, nucleus, and mitochondria [7]. Over the past few years, the functions of melatonin in plants have increased. One of the most important functions is that it acts as a protective molecule against biotic or abiotic coercion [8,9]. Thus, it is believed that melatonin may be effective in inducing stress tolerance in plants. Generally, as shown in Figure 1, starting from tryptophan, the pathway of melatonin biosynthesis varies among plant species and involves four enzymatic steps: first, the decarboxylation of tryptophan generates tryptamine, and then hydroxylation generates serotonin, or the hydroxylation of tryptophan first generates 5-hydroxytryptophan, and then decarboxylation generates serotonin. 5-hydroxytryptophan is either acetylated to *N*-acetylserotonin or methylated to 5-methoxytryptamine; these products are either methylated or acetylated to produce melatonin, respectively [10]. In previous reports, melatonin treatment enhanced the osmotic regulation of carbohydrates (trehalose) and amino acid (proline) molecules that normally increased to protect plants from high-temperature stress in maize (*Zea mays* L.) and kiwifruit (*Actinidia deliciosa*) seedlings [11,12], drought stress in *Carya cathayensis* and maize seedlings [13,14], and a combination of stresses (low temperature and drought) in rice plants [15]. Moreover, melatonin was shown to scavenge hydrogen peroxide in the aqueous phase at pH 7.0 and formed the AFMK (a main antioxidant metabolite of melatonin), and it was demonstrated that melatonin also had antioxidant properties [16].

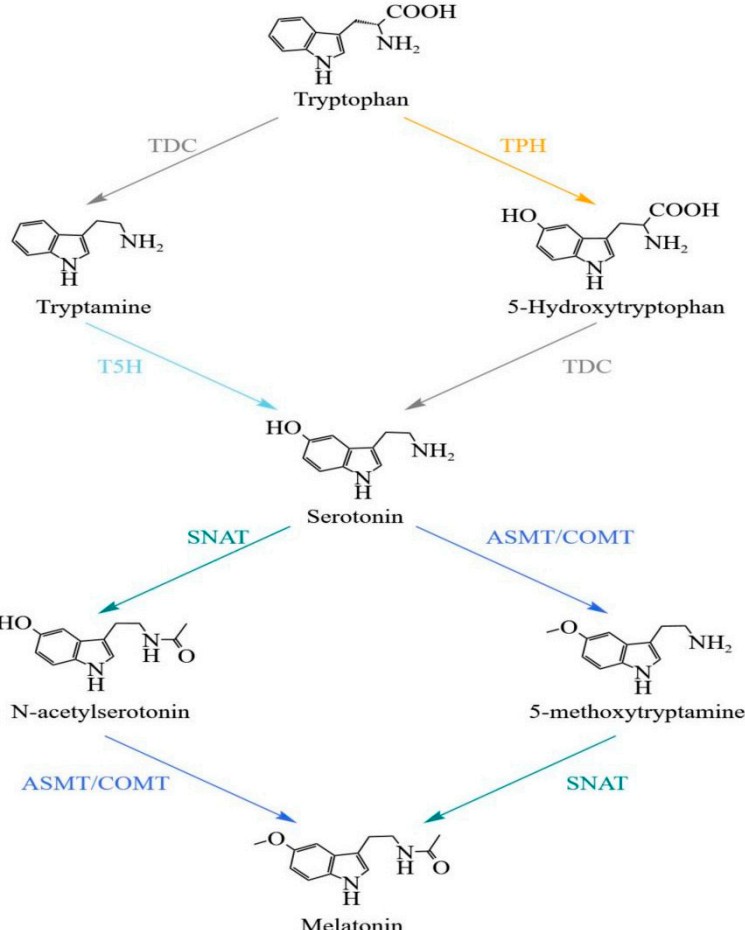

**Figure 1.** The routes for melatonin biosynthesis in plants. There are six enzymes involved in the biosynthesis pathway: TDC; TPH; T5H; SNAT; ASMT/COMT. (Revised from [17]).

In recent years, exogenously applied melatonin has been shown to improve the protective mechanism of plants against abiotic stress, which has attracted the attention of researchers around the world [7]. Some specific studies of melatonin-induced plant stress

resistance have demonstrated that melatonin can regulate the specific mechanism of response to abiotic stress. A detailed study on grafted *Carya cathayensis* plants revealed that the exogenous application of melatonin successfully restored plant growth and improved photosynthetic efficiency under drought stress [14]. Exogenous melatonin could reduce the negative effect of excessive light by improving the efficiency of the photosystem and rearranging the expression of chloroplast and nuclear coding genes in detached *A. thaliana* leaves [18]. Moreover, when pretreated with melatonin in *Citrullus lanatus* seedlings, it can alleviate NaCl-induced (300 mM) stress on the root by inhibiting stomatal closure and thereby protecting the photosynthesis apparatus [19]. Furthermore, melatonin could enhance the resistance to chilling injury by regulating the synthesis of polyamine in tobacco cells [20]. Genes involved in melatonin biosynthesis were involved in the response of *Vitis vinifera* suspension cells to high temperature and low temperature [21]. Nevertheless, although the positive role of melatonin in tolerating abiotic stress is similar, the role of melatonin in abiotic stress response still depends on the type of plant species, tissues, and abiotic stress.

The molecular mechanisms underlying stress-related gene expression and signal transduction are very complex, resulting from the synergistic effects of multiple genes and pathways. Hormones play a significant regulatory role in plant adaptation to stresses [22] and are involved in the early regulation of plant stress response through intricate interaction networks [23]. Several plant hormones, such as ethylene (ET), jasmonic acid (JA), abscisic acid (ABA), and salicylic acid (SA), are particularly prominent in abiotic stress-induced signaling pathways. Furthermore, in accordance with some findings, melatonin can interact with other plant hormones to further regulate different growth processes in hostile environments. Melatonin is considered a plant hormone; therefore, it can interact with other plant hormones (zeatin, JA, and ABA) to adjust the physiological processes [14]. Melatonin also modulates salinity via the SOS (Salt Overly Sensitive) pathway [24], which can modulate the abscisic acid (ABA) signal transduction pathway for regulating plant response to drought stress [14]. Fu et al. (2017) [25] showed that exogenously applied melatonin improved cold resistance by inducing endogenous melatonin production, and the study suggested that both ABA-independent and ABA-dependent pathways may contribute to melatonin enhanced cold resistance in *Elymus nutans*. Moreover, significant expression of rice ASMT-mRNA after treatment with abs*cis*ic acid and methyl jasmonic acid was observed, specifying that melatonin had a certain relationship with other hormones [26]. Nevertheless, more research is essential to clarify the interrelationship between melatonin and other hormone signal molecules to cope with stress.

Watermelon (*Citrullus Lanatus*), a major horticultural crop with a long cultivation history, has been found to originate from desert regions of Africa and therefore has the characteristic of drought resistance, which is an ideal material for studying the formation of a drought resistance mechanism. Due to the impact of the harsh environment in recent years, watermelon is susceptible to various abiotic stresses during growth and development [27]. Consequently, improving watermelon stress resistance is critical for ensuring growth and productivity. According to the latest research advances on melatonin, the results have demonstrated that melatonin has a positive role in dealing with adverse environments. Mannitol, as an osmolyte, is commonly used to simulate osmotic stress. In the current study, we focus herein on the osmotic stress response of melatonin in watermelon suspension cells, the alleviating impacts of exogenously applied melatonin on the osmotic stress response, as well as the genes involved in melatonin biosynthesis in watermelon. In addition, the signaling biosynthesis pathway of osmotic stress-induced melatonin in watermelon cells was explored by plant hormone signal-binding inhibitors. These results describe the interaction between melatonin and plant hormones to respond to adversity.

## 2. Materials and Methods

### 2.1. Plant Materials and Culture Conditions

Suspension cell cultures were established from calli induced from leaves; the method for maintaining the cell culture and the growth of the culture were detailed in our previous publication [28].

### 2.2. Measurement of Cells Growth Responses to Osmotic Stress

Packed cell volume (PCV) [29] is used as an indicator of cell growth in response to osmotic stress. In this study, after subculture, PCV was determined on the 7th day, D-mannitol was dissolved in MS medium, melatonin (Yuanye, Shanghai, China) was dissolved in 6% ethanol solution at a stock solution of 10 mM concentration, and the working concentration was 10 μM. To determine the impact of different osmotic strengths, various mannitol concentrations (0–400 mM) were added to cells at sub-cultivation. Various concentrations of melatonin (0, 1, 5, 10, 25, 50, and 100 μM) were supplied to cells under 100 mM mannitol to ensure the optimum concentration of melatonin to alleviate osmotic stress. After that, to investigate whether melatonin can relieve the inhibition of cell growth caused by osmotic stress, cells were pretreated with the optimum concentration of melatonin for 12 h, then treated with different mannitol concentrations during the cell growth phase (the fourth or fifth day of cell subculture). Evans blue stain [30] was used to quantify the cell viability when cells were treated with mannitol stress treatment, and the cells were imaged by a digital image acquisition system under an Axio Observer Z1 (Zeiss, Germany) using 10× objectives.

In order to speculate whether melatonin correlates with superior oxidative homeostasis in watermelon cells, malondialdehyde (MDA) and peroxidase (POD) were measured. Cells were grown under stress conditions (100 mM mannitol treatments for 1, 2, 4, 6, 8, and 12 h) with or without melatonin pre-treatment, and then aliquots (0.1 g) of filtered suspension cells were collected. Malondialdehyde kit and Peroxidase kit (Keming, Suzhou, China) were adopted to quantify the MDA content and POD activity. ROS was detected by DCFH-DA (10 μM) under osmotic stress on the basis of the instruction of the Reactive oxygen species assay kit (Solarbio, Beijing, China).

### 2.3. Identification and Sequence Analyses of OMT Genes

We downloaded the second edition of the watermelon database (watermelon_v2.pep) from Cucurbit Genomics Database (http://cucurbitgenomics.org/, accessed on 7 June 2022) [31]. In order to comprehensively identify *Citrullus lanatus OMT* family genes, the conserved motif and protein functional domain of reported ASMT and COMT proteins were analyzed by using NCBI's Conserved Domain Search (http://www.ncbi.nlm.nih.gov/Structure/cdd/wrpsb.cgi, accessed on 7 June 2022) and the PFAM website (http://pfam.xfam.org/, accessed on 7 June 2022) [32], and found that they all contained the *O*-methyl transferase domain (PF00891). The downloaded HMM model of ID PF00891 from the PFAM website was used as a reference sequence to conduct Hidden Markov Model (HMM) analysis in the watermelon protein database, and we selected $\text{e}^{-\text{value}} \leq 1 \times \text{e}^{-10}$ as a candidate OMT protein. We obtained candidate proteins through the SMART tool website (http://smart.embl-heidelberg.de/, accessed on 7 June 2022) [33] for protein sequence identification. Using TBtools1.0987 software, amino acid sequences of SNAT (AtSNAT) families of *Arabidopsis thaliana* were compared with the watermelon database under the parameter conditions of E-value: $1 \times \text{e}^{-10}$ and Number of Hits: 500. Combined with NCBI Blast analysis of website (https://www.ncbi.nlm.nih.gov/, accessed on 7 June 2022), we carried out screening of *ClSNAT*.

The isoelectric point (pI) and molecular weight (MW) of these identified *ClOMT* and *ClSNAT* genes were predicted by ExPASy (https://www.expasy.org/, accessed on 8 June 2022) [34]. To investigate the relationships among the *ClOMT* and *ClSNAT* genes, the phylogenetic trees of the melatonin biosynthesis-related genes *ClOMT* and *ClSNAT* were constructed based on the neighbor-joining method and 1000 replicated bootstrap values

using MEGA 7.0 software [35]. In addition, using the PlantCARE (http://bioinformatics.psb.ugent.be/webtools/plantcare/html/, accessed on 12 June 2022) [36] online databases, the regulatory elements in the 2000-bp genome sequence upstream of the encoding *ClOMT* and *ClSNAT* gene sequences were investigated.

### 2.4. Quantification of Gene Expression by qRT-PCR

*ClOMT* and *ClSNAT* genes involved in melatonin biosynthesis were identified to measure gene expression after different treatments. Taking an equal portion (40 mL) of suspension cell on the 5th day after subculture, cells were treated with 100 mM D-mannitol, 100 μM MeJA (methyl jasmonate,) 100 μM ABA (abscisic acid), and 100 μM SA (salicylic acid) (Sigma, Shanghai, China) for 1, 2, 4, 6, 8, and 12 h, respectively. Then the watermelon cells were filtered to remove supernatant and accurately weighed (0.5 g), and quickly put into liquid nitrogen. The preparation for hormones was as follows: MeJA was mixed with 0.25% ethanol solution, SA was dissolved with 37.5% ethanol solution to a stock solution of 100 mM, and ABA was dissolved in ultrapure water to a stock solution of 10 mM.

Ibuprofen (IBU, Sigma, Shanghai, China), a jasmonate biosynthesis inhibitor, was dissolved in DMSO, and 50 mM as a stock solution concentration. Fluridone (FLU), an ABA biosynthesis inhibitor, was dissolved in distilled water, and the stock solution was also 50 mM. Cells were pretreated with 50 μM IBU or 50 μM FLU for 30 min prior to the addition of 100 mM D-mannitol. The same procedure was used for IBU and FLU treatments.

Cells were ground with 2 steel balls by Tissue Lyser (Jingxin, Shanghai, China). Total RNA was extracted using the RNAprep Pure Plant Plus Kit (TIANGEN, Beijing, China), and total RNA was treated with DNAse in the process of the extraction; the concentration of RNA was measured by an Ultramicro spectrophotometer (Implen/ N50 Touch, Implen, Germany). Reverse transcription of the extracted RNA into cDNA using the PrimeScript™ RT reagent Kit with gDNA Eraser (Perfect Real Time) (Takara Bio, Otsu, Japan) was performed using the manufacturer's instructions.

Real-time quantitative PCR (RT-qPCR) was used to detect gene expression, and RT-qPCR was performed using a PikoReal 96 Real-Time PCR System (Thermo Fisher Scientific™, Waltham, MA, USA). The primers are listed in Table S1, designed by Primer Premier 7, which were amplified by PCR primers. The *β*-actin gene (Cla97C02G026960) of watermelon was used as the reference gene and was much more stable than the 18S rRNA. A total reaction volume of 10 μL was used. The reaction included 1 μL template, 5 μL ChamQ Universal SYBR qPCR Master Mix (Vazyme, Nanjing, China), 0.8 μL reverse primer, 0.8 μL forward primer, and 2.4 μL double distilled water to a total volume of 10 μL. PCR detection was performed under the following conditions: 95 °C for 7 min followed by 95 °C of 40 cycles for 5 s and 60 °C for 30 s [28]. ABI Step was used to carry out amplification and data analysis.

### 2.5. Quantification of Melatonin by HPLC

After taking an equal portion (40 mL) of suspension cells on the 5th day after subculture, cells were then treated with different reagents. Cells were treated with D-mannitol for 1, 2, 4, 6, 8, 12, and 24 h, different hormones (ABA, MeJA, and SA) for 24 h, and cells were pretreated with hormone inhibitors (IBU and FLU) for 6 h before adding mannitol for 24 h, the concentrations were consistent with the above. The watermelon cells were filtered to remove supernatant and accurately weighed (0.5 g), and quickly frozen in liquid nitrogen, then they were placed in a −80 °C refrigerator. A powder sample of watermelon cells was dissolved with 1 mL of 100% methanol, and the dissolved sample was then ultrasounded at room temperature for 10 min; the supernatants were collected after centrifugation (8000 rpm, 10 min) and filtered through a 0.22-μm disposable needle filter for LC-MS/MS detection. Melatonin standard solutions (Sigma, Shanghai, China, ≥98%) were dissolved in methanol (1 mg/mL) and diluted to concentrations of 1000, 500, 250, 125, 62.5, and 31.25 μg/L, respectively.

Quantification and confirmation of *Citrullus lanatus* melatonin were carried out on an Exactive[TM] plus orbitrap mass spectrometer coupled with Dionex UltiMate 3000 UPLC system (Thermo Fisher Scientific, USA). The cell extracts (2 μL) were separated on an Agilent InfinityLab EC-C18 column (2.1 × 100 mm, 2.7 μm particle size), and the mobile phase contained 100 (*v/v*) methanol at a flow rate of 0.3 mL/min, methanol and 0.1% HCOOH aqueous solution was used for washing column. The UV1 channel was set to 277 nm, the oven temperature was 35 °C, and the DAD detection range was 200 to 800 nm at a data collection rate of 5 Hz.

### 2.6. Statistical Analysis

The experiments were repeated three times using the independent samples. The data were statistically calculated by a one-side *t*-test ($p < 0.05$, $p < 0.01$). The expression of relative transcript detected by qPCR was calculated by the $2^{-\Delta\Delta Ct}$ method.

## 3. Results

### 3.1. Exogenous Melatonin Can Relieve Cell Growth Inhibition of Citrullus Lanatus Caused by Osmotic Stress

The possible role of melatonin as a growth promoter responding to osmotic stress was broadly noticed in various plant species [37,38]. However, the effects of melatonin on osmotic stress at the cellular level have not been reported. In the experiment, PCV was used as an indicator of cell growth promotion or inhibition; 100 mM mannitol inhibited approximately 20% of cell growth compared to normal cultured cells (blank control), and when cells were pretreated with melatonin, the growth was only inhibited about 10% (Figure 2A). Furthermore, when cell viability was determined, it could be demonstrated that cell death remained at significantly lower levels after melatonin pretreatment under 100 mM mannitol stress (Figure 2B). Exogenously applied melatonin could mitigate the inhibition of osmotic stress on cell proliferation and growth (Figure 2A,B). The optimum concentrations of melatonin for alleviating 100 mM mannitol stress were 10 μM (~88% of blank control), and the alleviation was significant when compared to the non-melatonin pretreated group. Nevertheless, both the higher and lower doses of melatonin were less effective in improving cell growth under osmotic stress; even inhibitory effects may exist at high concentrations of melatonin. Similar results have been reported in *B. juncea* [39], where it was observed that 0.1 μM melatonin concentration promoted root length, and at 100 μM concentrations, there was a significant inhibitory effect in 2-d-old, etiolated seedlings.

To investigate the impact of the application of melatonin on cell growth under different osmotic strengths, mannitol (0–400 mM) was added to the cell cultures. With the increase in mannitol concentration, cell growth decreased significantly (Figure 2C). The decrement was approximately 20% with 100 mM mannitol and nearly 50% with 400 mM mannitol when contrasted to blank control. Though the inhibition of cell growth was alleviated significantly by melatonin in all treatments, a most significant recovery ($p < 0.05$) was found in 400 mM mannitol treatment among different mannitol treatments (Figure 2C), suggesting that the protective effect of melatonin on cell growth under osmotic stress depends on stress intensity.

### 3.2. Melatonin Can Partially Alleviate the Cellular Oxidative Burst Triggered by Osmotic Stress

Melatonin has been found to be a broad-spectrum antioxidant with high ROS clearance capacity and efficiency [40,41]. Therefore, it is speculated whether the application of melatonin correlates with superior oxidative homeostasis in watermelon cells. We measured malondialdehyde (MDA) levels, peroxidase (POD) activity, and ROS in watermelon cells in our experiment to relate the previously observed positive effects of melatonin in cell growth parameters to its known antioxidant effects (Figure 3). One reading of membrane oxidative degradation is the lipid peroxidation level reported by the product malondialdehyde (MDA) [42]. MDA levels were followed to address the osmotic stress in cells at the onset of growth (day 4 or 5 after subcultivation). As the stress time increased, higher levels of

MDA were found correspondingly, and this increase was already manifested at 1 h after the onset of osmotic stress. When stressed for 12 h, the MDA level was more than 1.6 times in contrast to the blank control and significantly higher as compared to pre-treatment of melatonin. When pre-treated with melatonin, MDA levels also increased with time, while the increase of MDA was minor and almost remained at a steady-state level (Figure 3A). Thus, with respect to MDA levels, this demonstrates that melatonin is effective in inhibiting the production of MDA to maintain oxidative homeostasis.

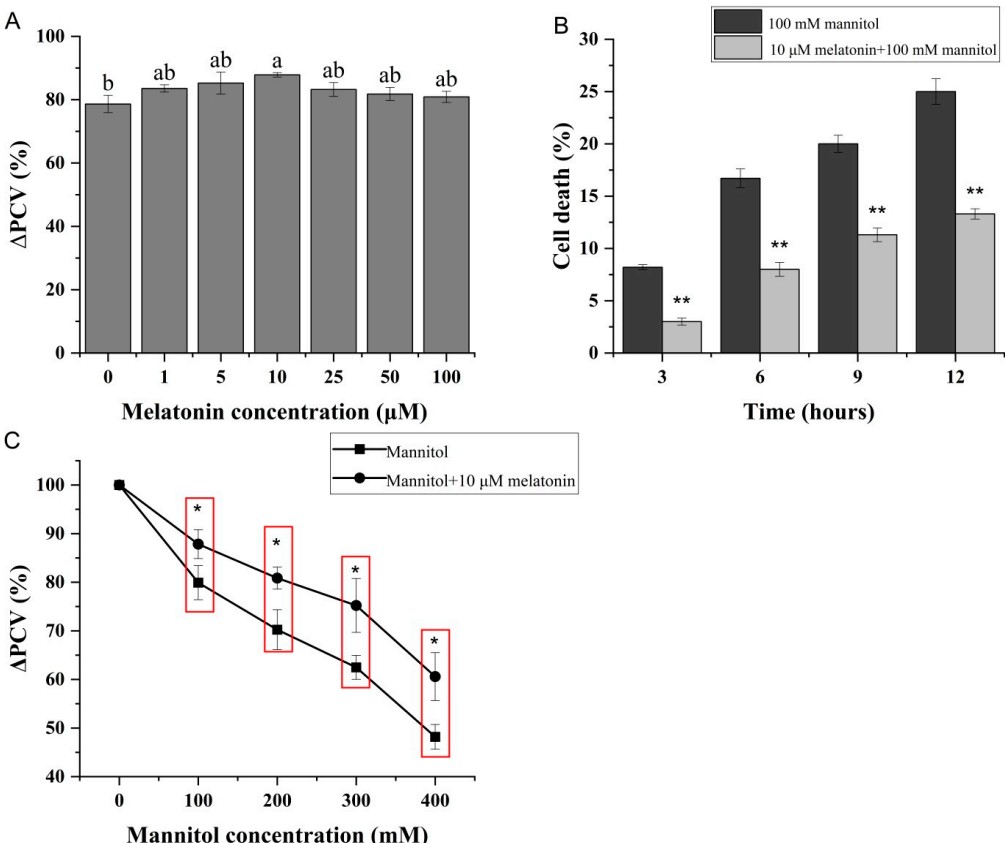

**Figure 2.** Response of *Citrullus Lanatus* cell growth treated with mannitol indicated by packed cell volume (ΔPCV). (**A**) Effects of exogenous melatonin with different concentrations on cell growth under 100 mM mannitol. The PCV of control was defined as 100%. (**B**) The relative ratio of dead cells was quantified at different mannitol stress times (3, 6, 9, and 12 h) with melatonin pretreatment or not. (**C**) Decrease of PCV caused by different mannitol concentrations can be alleviated by the pretreatment of melatonin. Each data point represents the average value from three independent experimental series. Different lowercase letters indicate the significance of each treatment ($p < 0.05$). * and ** indicate significant differences of $p < 0.05$ and $p < 0.01$ by a one-side *t*-test.

Peroxidase (POD) is one of the vital antioxidant enzymes in organisms that can be enhanced to improve plant tolerance. Consequently, steady-state levels of peroxide, along with the activity of peroxidase, were determined. In the non-melatonin pretreated cells, POD activity increased concomitantly with increasing stress time, and steady-state levels of peroxide rose accordingly. This increase was less acute in cells as compared to pretreated with melatonin. In melatonin-pretreated cells, the pattern was identical, except the cells maintained a high activity of POD, which correlated with a significant increase in peroxide steady-state levels. The difference was pronounced in 4 h-stress. Thus, pre-treatment of melatonin leads to a higher peroxidase activity under osmotic stress resulting in peroxide homeostasis (Figure 3B). In addition, the positive role of melatonin in quenching ROS production was directly measured. When cells were subjected to osmotic stress, ROS generated continuously, while in cells pretreated with melatonin, the generation of ROS

decreased compared to direct mannitol stress. Therefore, it could be directly seen that exogenous melatonin counteracted the increased ROS levels induced by osmotic stress (Figure 3C).

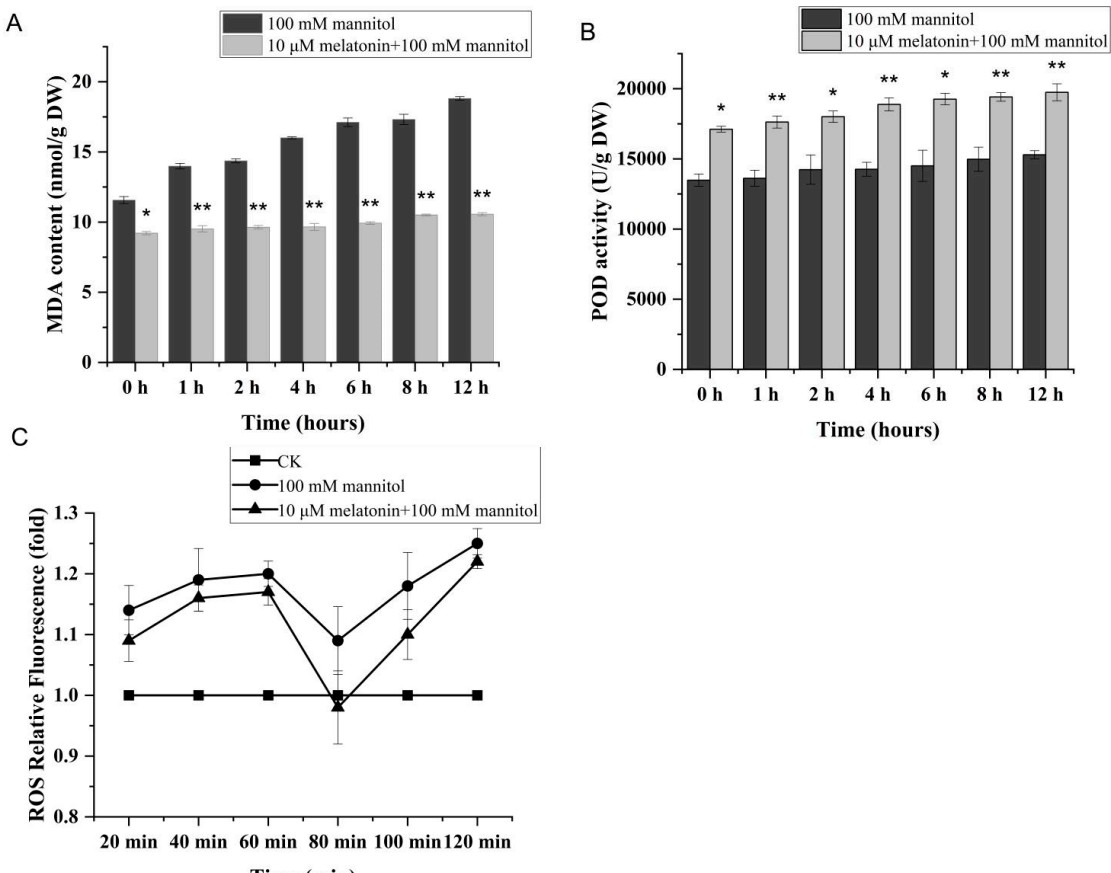

**Figure 3.** Oxidative stress of watermelon cells grown under stress conditions (100 mM mannitol treatments for 1 h, 2 h, 4 h, 6 h, 8 h, and 12 h) with or without melatonin pre-treatment. (**A**) Lipid peroxidation measured as malondialdehyde (MDA) content; (**B**) Antioxidant enzyme measured as peroxidase (POD) activity. (**C**) The production of ROS in response to the solvent control, 100 mM mannitol, and 10 mM melatonin + 100 mM mannitol treatment for different minutes (20, 40, 60, 80, 100, and 120 min). The control is normalized to 1. Each data point represents the average value from three independent measurements. Error bar indicates ± S.E. * and ** indicate significant differences by *t*-test at a confidence level of $p < 0.05$ and $p < 0.01$.

### 3.3. Sequence Analysis of ClOMT and ClSNAT Genes

Serotonin *N*-acetyltransferase (SNAT) and *N*-acetylserotonin methyltransferase (ASMT) are the penultimate and final-step enzymes in melatonin biosynthesis in plants, respectively, and they both had a rate-limiting effect [8,43]. Besides ASMT, caffeic acid *O*-methyltransferase (COMT) also had ASMT activity and could catalyze the production of melatonin [44,45]. According to the catalytic properties of enzymes, two enzymes are attributed to the *O*-methyltransferase (OMT) family. They were used to identify 16 candidate *ClOMT* genes and 2 candidate *ClSNAT* genes in the watermelon protein database. Sixteen *ClOMT* genes were unevenly distributed among four chromosomes, with chromosome 2 containing the most genes (nine genes), followed by chromosome 10 (five genes), while chromosomes 7 and 9 contained the fewest *ClOMT* genes (one gene for each); some *ClOMT* genes were distributed in adjacent regions of the same chromosome, and they were named *ClOMT1* to *ClOMT16* according to their chromosomal positions (Figure 4A). *ClSNAT1* and *ClSNAT2* were allocated on chromosomes 4 and 5, respectively (Figure 4B).

The number of amino acids encoded by *ClOMT12* was 897 aa, and the lengths of other *ClOMT* genes ranged from 1017 aa to 1113 aa. The analysis of physiochemical properties of ClOMT proteins (Table S2) showed that the MW of proteins varied from 73.84 to 92.05 kD, all 16 ClOMT proteins were hydrophobic acidic proteins, that is, the Grand average of hydropathicity (GRAVY) values were positive and the isoelectric points (PI) were between 5.01 and 5.13. Moreover, the lengths of *ClSNATs* were 744 aa and 576 aa, and the molecular weight of the proteins were 61.86 and 48.59 kD, which was less than *ClOMTs*. Consistent with ClOMT proteins, both ClSNAT proteins were hydrophobic acidic proteins (Table S2).

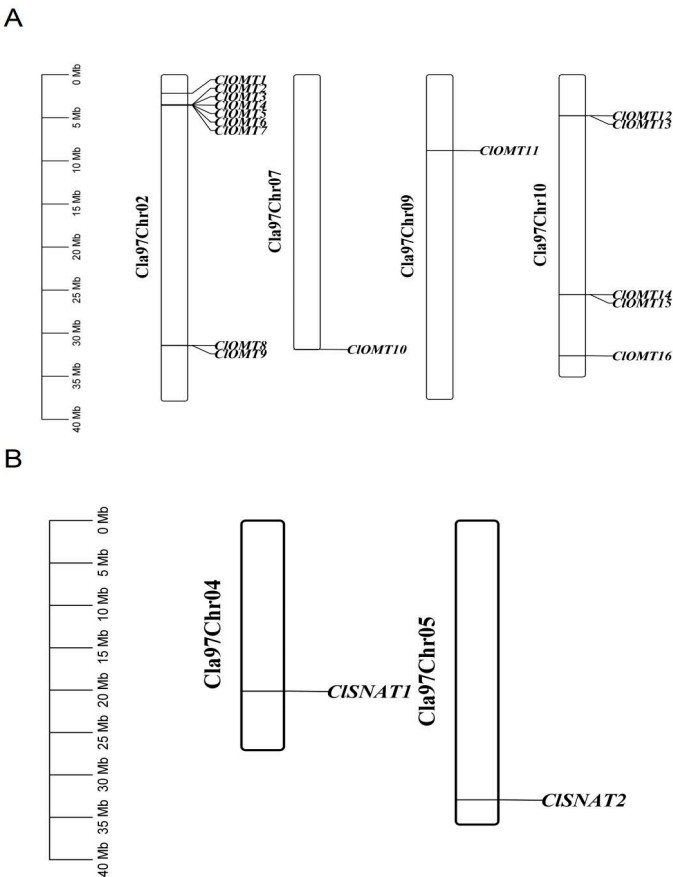

**Figure 4.** Chromosome location of *ClOMT* and *ClSNAT* genes in watermelon. (**A**) *ClOMTs*. (**B**) *ClSNATs*.

### 3.4. Phylogenetic Relationships and Cis-Regulatory Elements of ClOMTs and ClSNATs

According to the sequence alignment, by constructing a phylogenetic tree with a bootstrap value of 1000 by using MEGA 7.0, the evolutionary relationships among *C. lanatus*, *A. thaliana*, *O. sativa*, and *S. lycopersicum* were explored. According to the evolutionary relationship of OMT family members, 16 *ClOMT* genes were clustered into three subclasses. These three subclasses were named Groups I, II, and III. Group I consisted of four members (*ClOMT10*, *-12*, *-13*, and *-16*), while *ClOMT1*, *-11*, *-14*, and *-15* belonged to Group II, another eight identified genes of the ClOMT family comprised Group III (Figure 5A). Moreover, *ClOMT* genes in group III cluster together, while there were no aggregations of genes from other species. As for *ClSNATs*, two *ClSNAT* genes are also grouped into two clades (Figure 5B).

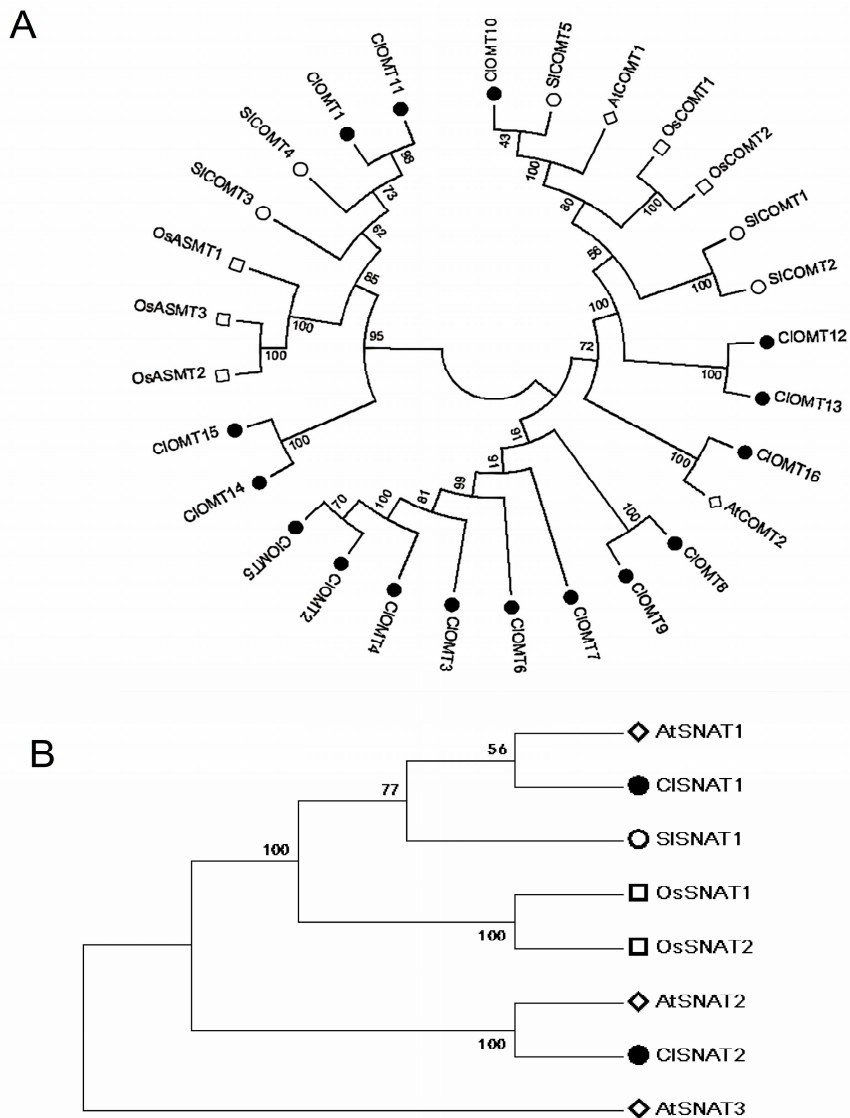

**Figure 5.** Phylogenetic analysis of the *OMT* (*ASMT* and *COMT*) and *SNAT* genes of watermelon (Cl, ●), Arabidopsis (At, ◇), rice (Os, □), and tomato (Sl, ○). (**A**) *OMT* (*ASMT* and *COMT*) genes. (**B**) *SNAT* genes.

Identification of *cis*-elements in the promoter sequences is necessary for understanding the regulation and function of genes. To investigate the *cis*-elements of the 16 *ClOMT* and 2 *ClSNAT* genes, 2000-bp of sequence upstream from the start codon was extracted to analyze the promoter region. Based on the functional annotation, the *cis*-elements of the *ClOMT* and *ClSNAT* genes were mainly classified into two categories, those involved with stress responses, as well as those that respond to plant hormones (Table S3). *Cis*-acting elements responding to abiotic stress mainly included anaerobic inducible *cis*-elements (ARE), low-temperature stress-responsive *cis*-elements (LTR), light-responsive *cis*-elements (G-box), drought stress-responsive *cis*-elements (MBS) and defense and stress-responsive *cis*-elements (TC-rich repeats). Also, phytohormone-responsive *cis*-elements were detected; methyl jasmonate responsiveness (TGACG-motif), abscisic acid responsiveness (ABRE), and salicylic acid responsiveness (TCA-element). These suggested that plants could regulate their adaptability to adversity through a complex signal regulatory network.

### 3.5. Osmotic Stress Induced ClOMT and ClSNAT Genes Expression and Melatonin Biosynthesis

To clarify the response of endogenous melatonin to osmotic stress, we measured the transcript level of *ClOMT* and *ClSNAT* genes and the accumulation of melatonin after treatment with mannitol. The expression of *ClOMTs* and *ClSNATs* exhibited an upregulated expression contrasted with the blank control (Figure 6A). From bioinformatics, it can be seen that among the 16 ClOMT proteins, ClOMT10 is annotated as caffeic acid 3-*O*-methyltransferase 1, and ClOMT12 and ClOMT13 are annotated as caffeic acid 3-*O*-methyltransferase-like (Table S2). By comparing and analyzing *ClOMTs* sequence similarity with *COMTs* of melatonin biosynthesis genes reported in other plant species, it could be concluded that *ClOMT10*, *ClOMT12*, and *ClOMT13* had higher similarity than other *ClOMT* genes (Table S4), and they can all be cataloged Group I with *ClOMT16* according to the phylogenetic tree. It could also be inferred that *ClOMT10*, *ClOMT12*, *ClOMT13*, and *ClOMT16* were genes involved in melatonin synthesis [46]. The results showed that four *ClOMT* genes were differentially expressed under different stress times (Figure 6A). *ClOMT12* and *ClOMT16* showed significant changes in expression under stress treatments, and *ClOMT12* was most significantly upregulated under 1 h stress and *ClOMT16* under 6 h stress. While *ClOMT10* and *ClOMT13* showed downregulated expression all the time. As for *ClSNATs*, it could be concluded that *ClSNAT1* was hardly expressed, while *ClSNAT2* was upregulated at 2 h treatment until it reached its maximum expression at 8 h (Figure 6A). Concerning *ClOMTs* and *ClSNATs*, which encode two enzymes involved in melatonin accumulation, some were upregulated in the early phase of osmotic stress, which is conducive to melatonin accumulation, and melatonin was significantly accumulated under osmotic stress (Figure 6B). The outcomes demonstrated that the endogenous level of melatonin in watermelon cells could be induced by osmotic stress, and genes might play a role in the osmotic stress resistance of watermelons.

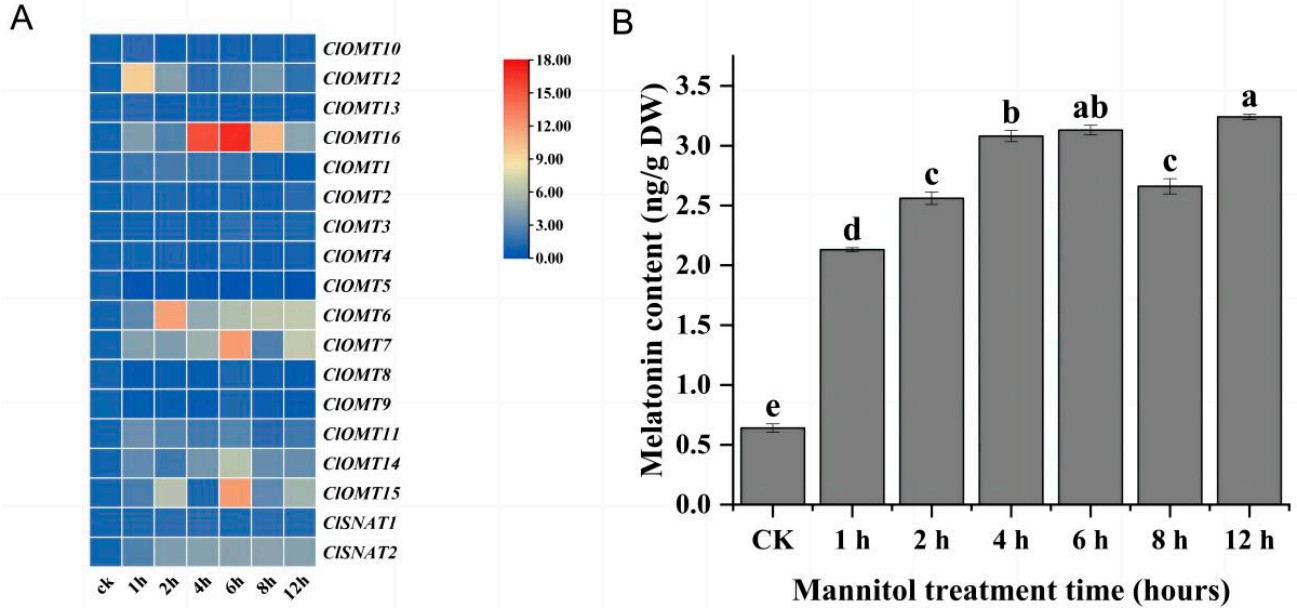

**Figure 6.** The relative expression of *ClOMTs* and *ClSNATs* and melatonin content in watermelon cells grown under stress conditions (100 mM mannitol treatments for 1 h, 2 h, 4 h, 6 h, 8 h, and 12 h). (**A**) The relative expression levels of *ClOMTs* and *ClSNATs*; (**B**) The melatonin contents were determined by UPLC-MS. The cells were treated with MS medium as a blank control. Each data point represents the average value from three independent experimental series. Different lowercase letters in the picture represent significant differences between different treatments ($p < 0.05$) according to the *t*-test. Error bar indicates ± S.E.

### 3.6. Exogenous Signal Molecules Can Enhance ClOMT and ClSNAT Genes Expression and Melatonin Biosynthesis

*Cis*-acting elements in the promoter regions concerning *ClOMTs* and *ClSNATs* showed that they contain a series of hormone signal molecules to respond to osmotic stress, embodying MeJA, SA, and ABA (Table S3). However, ethylene-responsive *cis*-element, which was used as a stress hormone and widely exists in plants, was not identified in the promoter regions of *ClOMTs* and *ClSNATs*.

To gain insights into the response of melatonin to plant hormones, exogenous MeJA, SA, and ABA were applied to measure changes in *ClOMT* and *ClSNAT* gene expression and melatonin content. Compared with blank control, the expression of *ClOMT10*, *ClOMT12*, *ClOMT13,* and *ClOMT16*, as well as *ClSNAT2*, exhibited specifically higher expression at certain time points after the addition of ABA, MeJA, and SA. Four *ClOMTs* reached the highest value at 8 h or 12 h with ABA treatment, but for MeJA and SA treatments, there is no obvious pattern in the expression of four *ClOMTs* (Figure 7A). For *ClSNATs*, we can observe that *ClSNAT2* had a marked expression more than *ClSNAT1* under three hormone treatments, and *ClSNAT1* showed low expression all the time under various hormone treatments. In addition, among three hormone treatments, *ClSNAT2* had the highest expression in ABA treatment (Figure 7B), but intriguingly, *ClSNAT2* has no phytohormone-responsive *cis*-elements. Melatonin was significantly accumulated after being treated by ABA, MeJA, and SA compared to the blank control. However, contrasted with the 100 mM mannitol treatment, only the ABA treatment induced an analogical increase in melatonin content, while the other two treatments only caused minor accumulation (Figure 7C). It could be found that genes encoding melatonin biosynthetic enzymes are induced to a maximum extent under ABA treatment, thereby reinforcing the role of ABA in regulating melatonin levels under osmotic stress.

### 3.7. Effects of Inhibitors on Expression of ClOMT and ClSNAT Genes and Accumulation of Melatonin

Previous outcomes indicated that hormone signals might possess a crucial role in melatonin biosynthesis. To confirm this speculation, cells were pretreated with a jasmonate acid synthesis inhibitor (IBU) and abs*cis*ic acid synthesis inhibitor (FLU) to inhibit the endogenous biosynthesis of MeJA and ABA. As a result, the inhibitors reduced the expression of *ClOMTs* and *ClSNATs*; there were still *ClOMT12* and *ClOMT16* with weak expression. (Figure 8A,B). Interestingly, the expression of *ClOMT12* under the jasmonate acid synthesis inhibitor treatment was higher than that under the MeJA treatment. As presented in Figure 6C, both jasmonate acid synthesis inhibitor IBU and abs*cis*ic acid synthesis inhibitor FLU could significantly pull down the accumulation of melatonin, but cells with mannitol stress were still able to increase the synthesis of melatonin even pretreated with inhibitors (Figure 8C). Corresponding to this, the exogenous applied ABA and MeJA significantly enhanced melatonin accumulation (Figure 7C), and FLU observably inhibited the accumulation of melatonin. Hence, ABA was likely to play a dominant role in the regulation of melatonin levels under osmotic stress. It could also be inferred that hormone synthesis inhibitors inhibited the biosynthesis of endogenous hormones but were not completely suppressed; the hormones originally existing in cells still function when the cells encounter coercion.

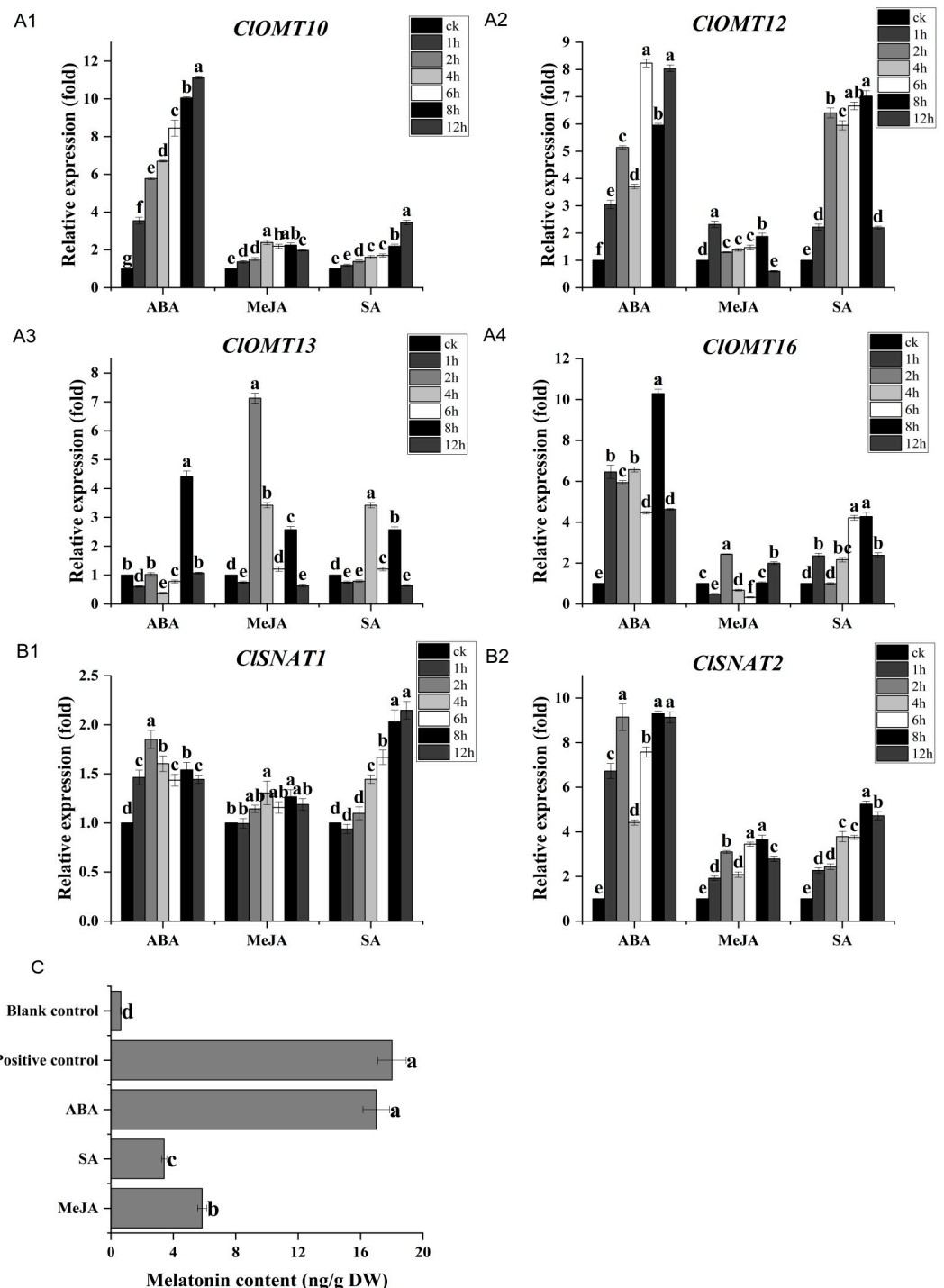

**Figure 7.** The relative expression of *ClOMTs* and *ClSNATs* and melatonin content in watermelon cells grown under various hormone treatments (1 h, 2 h, 4 h, 6 h, 8 h, and 12 h). (**A**: **A1–A4**) The expression patterns of *ClOMTs*; (**B**: **B1,B2**) The relative expression of *ClSNATs*; (**C**) The melatonin contents were determined by UPLC-MS with 100 μM ABA, 100 μM MeJA, and 100 μM SA for 24 h. The cells were treated with 100 mM mannitol as a positive control and MS medium as a blank control. Each data point represents the average value from three independent experimental series. Error bar indicates ± S.E. Different lowercase letters in the picture represent significant differences between different treatments ($p < 0.05$) according to the *t*-test.

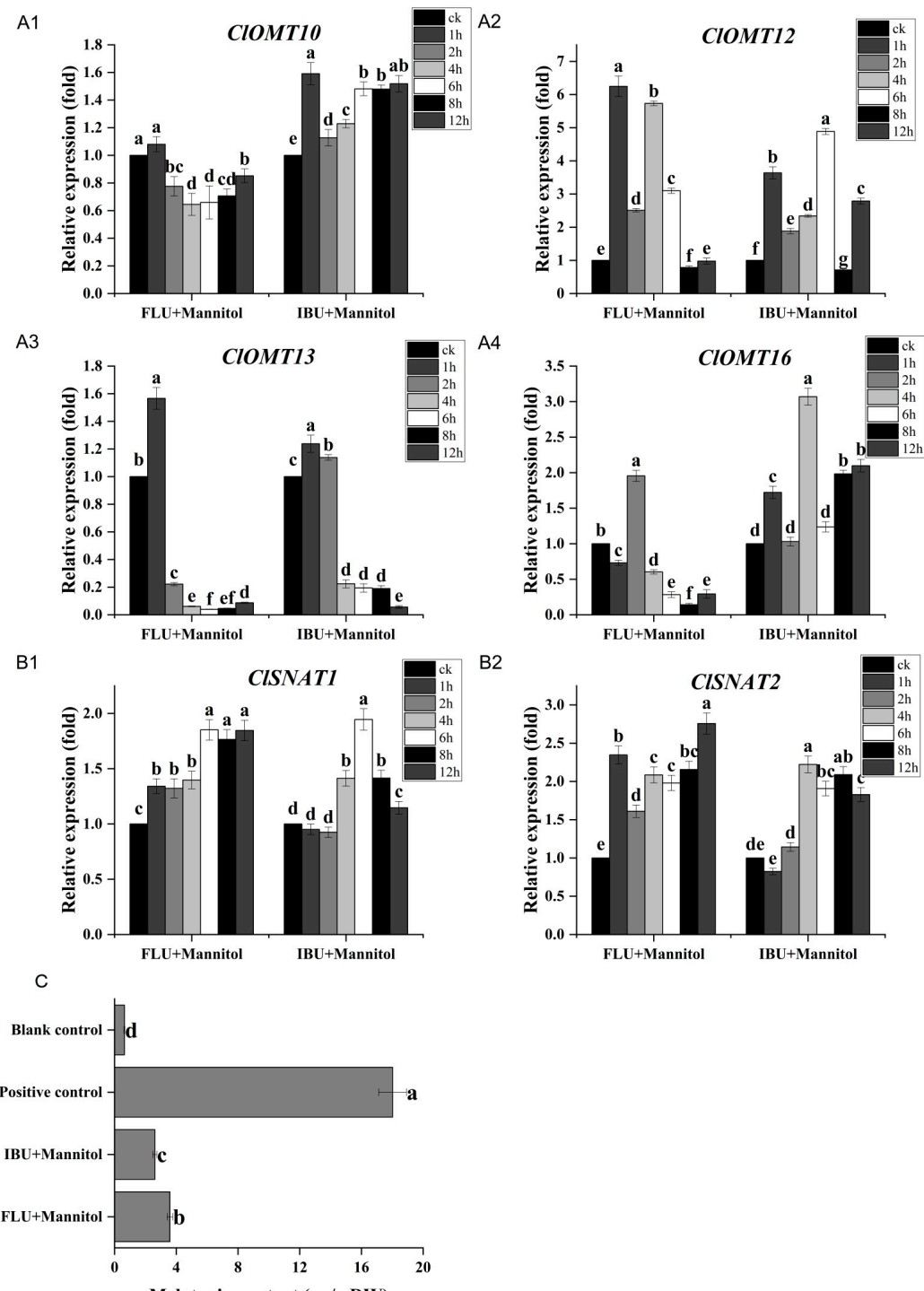

**Figure 8.** The relative expression of *ClOMTs* and *ClSNATs* and melatonin content in watermelon cells grown under various hormone inhibitor treatments were determined. (**A**: **A1**–**A4**) The relative expression of *ClOMTs*. The cells were pretreated with 50 μM FLU and 100 μM IBU for 30 min before adding 100 mM mannitol for 1 h, 2 h, 4 h, 6 h, 8 h, and 12 h; (**B**: **B1**,**B2**) The expression patterns of *ClSNATs*. The same treatments as above; (**C**) The melatonin contents were determined by UPLC-MS. The cells were treated with 50 μM FLU and 100 μM IBU for 6 h before adding 100 mM mannitol for 24 h. The cells were treated with 100 mM mannitol as a positive control and MS medium as a blank control. Each data point represents the average value from three independent experimental series. Error bar indicates ± S.E. Different lowercase letters in the picture represent significant differences between different treatments ($p < 0.05$) according to the *t*-test.

## 4. Discussion

To compensate for the lack of mobility, plants must produce a wide range of intricate defense systems to resist abiotic stress, such as drought, salinity, and extreme temperatures [47]. Melatonin occurs in different plant tissues, and melatonin biosynthesis results in the improvement of the critical function of plants' survival under diverse stress [7]. Elucidating the stress perception, signal, and response mechanisms of plants is crucial to understanding how they can improve stress tolerance under harsh environmental conditions [48]. In this study, with established watermelon suspension cells, it has been discovered that osmotic stress induced cell growth inhibition which can be mitigated by the pre-treatment of exogenous melatonin, and melatonin accumulation induced by osmotic stress may be bound up with hormone signals.

Melatonin, as a plant growth regulator, had a significant effect in cell culture [7]. Under osmotic stress, the synthesis and accumulation of melatonin can be rapidly activated and upregulated [49,50]. Nevertheless, studies have manifested that high levels of melatonin may be deleterious to plants and may cause growth suppression under special situations [51]. In the present study, pretreatment with melatonin modulated the proliferation and growth of watermelon cells; 10 μM melatonin showed maximum protection against osmotic stress. Moreover, it could be found that cell growth was inhibited by different strengths of mannitol-induced osmotic stress, whereas the application of exogenous melatonin to watermelon cells can significantly mitigate the inhibition (Figure 2). Interestingly, it was also observed that the promotion of cell growth was regulated by low melatonin concentrations but might not be regulated by high melatonin concentrations, even inhibited cell growth (Figure 2A).

Numerous studies have suggested exogenous applied melatonin affected the antioxidant capacity of plants after osmotic stress. Melatonin is associated with redox homeostasis, which is owing to the expression of antioxidant enzymes. For instance, in tall fescue (*Festuca arundinacea* Schreb.), it has been found that melatonin effectively improved the activity of antioxidant enzymes under high temperatures, thus promoting plant growth [52]. In naked oat (*Avena nuda* L.), it was shown that exogenous melatonin decreased $H_2O_2$ levels and enhanced the activities of SOD, POD, and CAT under drought stress [53]. SOD and POD are fully studied antioxidant enzymes and have been proven to be helpful in regulating the balance of oxidative reactions in resistance to osmotic stress. Current results show evidence that exogenously applied melatonin can act as a signaling molecule at the cellular level to induce the increase of the representative antioxidant enzyme POD as mannitol treatment time increases (Figure 3B), which increases its efficiency as an antioxidant. Exogenous applied melatonin could induce the decrease of MDA content, and MDA levels in different osmotic stress times were similar, as well as at 0 h (Figure 3A). In addition, when pretreated with melatonin, lower ROS levels occurred significantly in cells when the cells were under osmotic stress (Figure 3C). These all indicated that relief of the oxidative stress generated during osmotic stress is dependent on the presence of melatonin, which acted as a signal for the development of systemic acquired resistance [54].

Conversely, the increase of ROS levels in plants in response to external stimuli leads to an increase in melatonin levels, consequently activating antioxidant enzyme activity [55]. The study manifested changes in endogenous melatonin levels (Figure 6) resulting from the upregulation of *ClOMT* and *ClSNAT* genes. Melatonin works in synergy with antioxidant enzymes to remove ROS, enhance plants' antioxidant capacity, reduce the oxidative stress of cells, and protect plants from adverse environments [56]. Therefore, increasing endogenous melatonin can promote POD activity, and prompt the timely removal of excess ROS, thus enhancing watermelon's defense against osmotic stress.

The *COMT*/*ASMT* and *SNAT* genes were demonstrated to act as the rate-limiting genes for melatonin biosynthesis, which have been identified by sequence homologous alignment in a large number of plants. The *cis*-regulatory elements (CREs) in genes manifested that they can respond to hormones and stress. The levels and patterns of gene expression hinge on the existence or absence of CREs in their promoter regions [57]. In wa-

termelon, the melatonin biosynthesis genes also embody the elements responsive to stresses and hormones, such as MeJA-responsiveness, ABA-responsiveness, and SA-responsiveness (Table S3). The diverse CREs informed that melatonin biosynthesis was concerned with intricate multi-genes and signal cascade transduction networks.

Plants have evolved an intricate network of signaling molecules that sense and respond to osmotic stress. The phytohormones, including ABA, SA, and MeJA, can act as signaling molecules at the cellular level and induce a number of gene expressions under osmotic stress. Here, exogenous ABA treatment could notably enhance the expression of four *ClOMTs* and *ClSNAT2*, as well as prompt an increase in melatonin content comparable to the mannitol treatment (Figure 7). Also, when endogenous ABA synthesis was suppressed, both gene expression and melatonin synthesis were suppressed (Figure 8). Furthermore, exogenous applied MeJA and SA also increased the accumulation of melatonin and the expression of four *ClOMTs* and *ClSNAT2*, but lower than ABA treatment (Figure 7). Even though the MeJA synthesis was inhibited by IBU, melatonin content was still increased (Figure 8), and interestingly, the expression of *ClOMT12* under JA synthesis inhibitor treatment was higher than that under MeJA treatment. These findings indicated that hormones affected endogenous melatonin levels for signal transduction pathways involved in stress resistance, and ABA plays a dominant role in the expression of *ClOMTs* and *ClSNATs* when suffering osmotic stress. As for *ClSNATs*, intriguingly, there were no hormone *cis*-elements in *ClSNAT2*, whereas its expression was higher than that of *ClSNAT1* with hormone *cis*-elements (Figure 7B and Table S3). An explanation is gene expression depends not only on the existence of *cis*-elements but also on how and when the *trans*-acting elements interact with appropriate elements [57].

Interestingly, melatonin biosynthesis and its intermediate products at various subcellular sites depend on the order of enzyme reactions involved in melatonin synthesis [58]. When the last enzyme is SNAT, melatonin biosynthesis occurs in chloroplasts, whereas ASMT/COMT is involved in the terminal reaction in the cytoplasm [11,58]. The studies demonstrated that all plant SNATs were translocated in chloroplasts because they contained chloroplast transit peptides at their *N*-termini, and COMTs were only expressed in the cytoplasm due to a lack of transit peptides [59]. Now that the watermelon cells were cultured in the dark, in which chloroplasts were not formed or missing, consequently, we speculated that the formation of *N*-acetylserotonin by SNAT may not only depend on chloroplasts but ultimately lead to the synthesis of melatonin by ASMT/COMT in the cytoplasm. Therefore, the existence of a complex mechanism in the melatonin synthesis pathway requires further research.

By binding to specific hormone receptors and integrating them into the cellular metabolism, short-term and long-term responses are observed in target cells after perceiving the signal. This signal recognition ultimately leads to adaptation to adversity during growth and development and ensures the maintenance of cell integrity [60]. The beneficial role of melatonin in relieving stress can be widely attributed to improving cellular redox homeostasis and alleviating oxidative stress, as well as regulating the expression of stress response genes involved in signal transduction [50,61,62]. Our study found that the higher expression of antioxidant enzymes might lead to a higher tolerance to osmotic stress by activating melatonin metabolism, and it was proven that exogenous melatonin could alleviate the osmotic stress of watermelon suspension cells. Moreover, in higher plants, signaling molecules, such as ABA, SA, and MeJA, play significant roles in activating plant defense mechanisms in responding to osmotic stress [63,64]. With the watermelon suspension cell line, our study suggested that the involvement of ABA, MeJA, and SA in watermelon altered melatonin synthesis gene expression and that melatonin can interact with ABA, MeJA, and SA to modulate defensive responses. In addition, ABA signaling plays a dominant role in melatonin biosynthesis under osmotic stress. Hence, we proposed a pathway diagram for watermelon cells to regulate melatonin synthesis and metabolic pathways in response to osmotic stress (Figure 9). Ultimately, melatonin and other plant hormones are potentially important molecules to regulate plant growth and development

during osmotic stress, but there are still a good many aspects to elucidate about how they interact and how the tradeoff occurs when plants suffer osmotic stress.

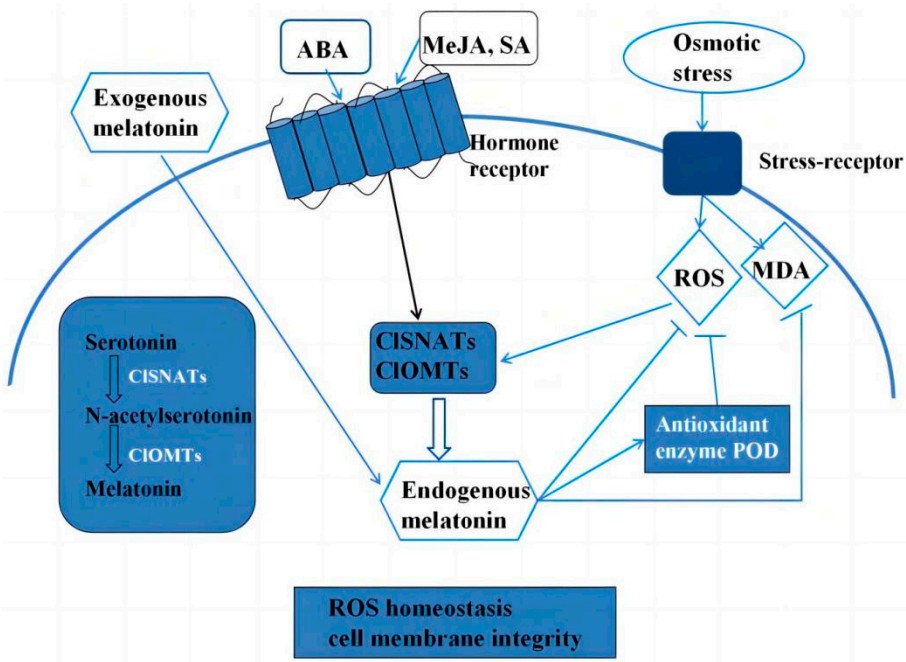

**Figure 9.** A hypothetical model for the regulation of melatonin biosynthesis against osmotic stress. "→" represents the stimulation, while "⊣" stands for the inhibition. "⇨" represents the synthesis pathway.

**Supplementary Materials:** The following supporting information can be downloaded at https://www.mdpi.com/article/10.3390/horticulturae9080927/s1, Table S1: Primer sequences for quantitative RT-PCR; Table S2: Physical properties of translated proteins of genes; Table S3: Prediction of *cis*-acting regulatory elements of *ClOMT* and *ClSNAT* genes; Table S4: The similarities of *ClOMTs* to *COMT* genes from other plants.

**Author Contributions:** Conceptualization, X.J., F.Q. and M.Y.; Methodology, M.Y. and M.L.; Data curation and Software, M.Y., M.L. and Z.D.; Validation, M.Y. and F.Q.; Formal analysis, M.Y. and F.Q.; Investigation, M.Y., M.L. and Z.D.; Writing-original draft preparation, M.Y.; Writing-review and editing, X.J. and F.Q.; Visualization, X.J. and F.Q.; Supervision, X.J. and F.Q.; Funding acquisition, X.J. All authors have read and agreed to the published version of the manuscript.

**Funding:** This work was financially supported by the National Natural Science Foundation of China (No. 32260801) and the Hainan Provincial Natural Science Foundation of China (No. 321RC473).

**Data Availability Statement:** The data that supports the findings of this study are available in the article.

**Acknowledgments:** We thank the technique assistance from the innovation and utilization team of tropical melon crop genetic germplasm, Hainan University.

**Conflicts of Interest:** The authors declare no conflict of interest.

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
