# Peer review of "Plant Hormone Signals Mediate Melatonin Synthesis to Enhance Osmotic Stress Tolerance in Watermelon Cells"

_horticulturae, doi:10.3390/horticulturae9080927_

Round 1
Reviewer 1 Report
The manuscript is well written with clearly presented results and descriptions. Some minor issues listed below:
1. Need some background on why and how mannitol induces osmosis stress.
2. Figure 3C. Any explanation on why ROS dropped at 80 min?
3. Figure 4A. what is the colorscale represented? relative expression level or gene counts or something else, please add labels or figure legends.
Reviewer 2 Report
In this manuscript, authors observed that exogenous application of melatonin to watermelon cells improve cell growth in response to osmotic stress by regulating antioxidant machinery of cells. In addition, treatment with different phytohormones as ABA, MeJA and SA revealed a response of gene expression up regulation of melatonin synthesis more related to ABA, which was comparable regarding osmotic stress. In general, the research seemed well conducted, however some issues need to be clarified or corrected.
- In line 191. Have the authors really measured mRNA concentration?
- About qPCR analysis. Total RNA was treated with DNAse to avoid genomic DNA contamination? This is indispensable to ensure that the results truly represent transcripts.
- Apparently, a single annealing temperature (60°C) for all primer pairs was used. Question: Have the authors tested previously what would be the best annealing temperature for each primer pair using a temperature gradient?
- A single β-actin gene (Cla97C02G026960) of watermelon was used as reference gene. However, no information was given about expression stability of this gene under the tested conditions. Question: Have the authors evaluated the β-actin expression “stability” under the tested conditions? This is crucial to ensure that precise transcript quantification.
- In table S4 and in the text it is not correct to refer % values of homology. Instead this use % of similarities.
- Add the name of groups in Figure S2A.
- Figure 7 need to be cited and explored in the text of discussion.
- The English language need to be extensively revised. Several type and grammatical errors are detected on the text.
Extensive editing of English language required
Reviewer 3 Report
Review of the article "Plant Hormone Signals Mediate Melatonin Synthesis to En- 2
hance Osmotic Stress Tolerance in Watermelon Cells" In the journal "Horticulture”.
The research topic is relevant.
The article contributes to the development of the theory and practice of plant stress tolerance. The article will be useful for physiologists, geneticists and other scientists.
The material presented by the authors is new and original.
Bibliography/References -good.
The discussions and conclusions are sound and supported by data.
The title corresponds to the material presented in the manuscript
The abstract is satisfactory.
Introduction: good.
However, it is recommended that the article be improved:
Methodology:
1. how was PCV determined? Please provide a reference to the methodology
2. Provide a reference to the Evans staining methodology
3. line 129 - provide reference to the methodology
4. line 197 - the primers used should be listed..
Results:
when describing sections 3.3, 3.4 and 3.5, references to figures are given, which should be cited in the text of the paper.
Discussion:
Good, but, in my opinion, it is incorrect to provide references to figures that have already been discussed in the Results section.The statistical analysis is convincing
The statistical analysis is convincing
Once the deficiencies have been corrected, the article will be ranked by Horticulture and, after a second review, may be recommended for publication.
Reviewer 4 Report
Manuscript ID: horticulturae-2532639
Type: Article
Title: Plant Hormone Signals Mediate Melatonin Synthesis to Enhance Osmotic Stress Tolerance in Watermelon Cells
Recommendation: Minor Revision
Here, the crucial role/function of melatonin in response to abiotic stress has been reported by the authors. The mechanistic approach after the exogenous application of melatonin to watermelon cells can has been shown to improve the growth responsiveness of watermelon to osmotic stress by regulating antioxidant machinery of cells. Overall, the manuscript is good related to the journal.
1. The role of melatonin should be more elaborated. Authors are requested to focus only on the ameliorative potential of the signalling molecule.
2. What was the reason of choosing watermelon as model plant in this study?
3. The title is good and updated.
4. Abstract is good and nicely written.
5. This article is written very well and may be helpful for readers/agronomists around the globe.
6. Plant Materials and Culture Conditions…Can the authors elaborate this section so that readers can easily employed their methods. It is only a suggestion.
7. The Introduction section is written very well. The authors have tried to elaborate the schematic pathways for melanin biosynthesis.
8. The material and methods section has described in detailed. Each and every section has a detailed and up to date account.
9. Results and discussion are appropriately discussed. The overall manuscript is good. However, there are some errors have been observed which is needed to improve before publication of the article.
10. Figures are nicely presented and statistically analysed.
11. Conclusion section is good. However, I think it needs revision/modifications. It seems like the authors have pasted the contents from the abstracts section. I am sorry so for these comments. But the conclusion section needs refining before the final publication of this article.

Round 2
Reviewer 3 Report
The article has certainly been significantly improved and in my opinion can be recommended for publication.